# Silicon Vacancy in Boron-Doped Nanodiamonds for Optical Temperature Sensing

**DOI:** 10.3390/ma16175942

**Published:** 2023-08-30

**Authors:** Masfer Alkahtani

**Affiliations:** Future Energy Technologies Institute, King Abdulaziz City for Science and Technology (KACST), Riyadh 11442, Saudi Arabia; mqhtani@kacst.edu.sa; Tel.: +966-553322891

**Keywords:** boron-doped nanodiamonds, temperature, silicon vacancy, temperature sensing

## Abstract

Boron-doped nanodiamonds (BNDs) have recently shown a promising potential in hyperthermia and thermoablation therapy, especially in heating tumor cells. To remotely monitor eigen temperature during such operations, diamond color centers have shown a sensitive optical temperature sensing. Nitrogen-vacancy (NV) color center in diamonds have shown the best sensitivity in nanothermometry; however, spin manipulation of the NV center with green laser and microwave-frequency excitations is still a huge challenge for biological applications. Silicon-vacancy (SiV) color center in nano/bulk diamonds has shown a great potential to be a good replacement of the NV center in diamond as it can be excited and detected within the biological transparency window and its thermometry operations depends only on its zero-phonon line (ZPL) shift as a function of temperature changes. In this work, BNDs were carefully etched on smooth diamond nanocrystals’ sharp edges and implanted with silicon for optical temperature sensing. Optical temperature sensing using SiV color centers in BNDs was performed over a small range of temperature within the biological temperature window (296–308 K) with an excellent sensitivity of 0.2 K in 10 s integration time. These results indicate that there are likely to be better application of more biocompatible BNDs in hyperthermia and thermoablation therapy using a biocompatible diamond color center.

## 1. Introduction

Over the past decades, unique optical and spin properties of color centers in fluorescent bulk/nanodiamonds have attracted a special interest in many promising applications including quantum communication [1,2], quantum photonics [3,4], and quantum sensing in biological sciences [5,6,7,8]. For more demanding and advanced applications of fluorescent nanodiamonds in biology, it was recently shown that boron-doped nanodiamonds (BNDs) were successfully implemented in local heating of tumor cells, when excited with near-infrared (NIR) lasers. This is an important and promising application for BNDs in hyperthermia and thermoablation therapy [9]. It was also recently shown that the NV color center in lightly boron-doped nanodiamonds (BNDs) can optically sense small temperature changes when heated with an 800 nm NIR laser with good sensitivity, which could be a better way to precisely monitor local temperature changes during tumor cell treatment [10]. The NV color center in diamonds has demonstrated the highest sensitivity in nanothermometry with nitrogen-vacancy (NV) centers due to its quantum property of electron spin with long coherence time, high photostability, and easy optical pumping [11,12]. The spin-dependent properties of the NV center can be optically initiated and then manipulated using optical and microwave-frequency techniques [13,14], which enable sensitive detection of electric, magnetic fields as well as small fluctuations in temperature at a nanoscale system [10,13,15]. However, spin manipulation of the NV center to conduct temperature sensing requires green light excitation and microwave frequency, which can potentially cause heating and photodamage of living cells and tissues, as well as autofluorescence that decreases optical temperature sensing sensitivity in hyperthermia and thermoablation therapy [16].

As the spin manipulation of the NV center to perform optical temperature sensing requires green light excitation and microwave frequency, which can potentially cause unwanted heating that could complicate precise optical temperature sensing during hyperthermia and thermoablation therapy, it is very important to introduce a more biocompatible diamond color center that can be excited and detected within the first biological transparency window. For this purpose, SiV color center in diamond can be excited with more red-shifted light excitations and detected within the biological transparency window where photodamage and overheating are greatly minimized [5,17]. The silicon-vacancy (SiV) defect in diamonds has been reported to have good optical temperature sensing within the biological transparency window by [15], with a temperature sensitivity reaching 360 mk/$\sqrt{\mathrm{Hz}}$ in bulk diamond and 521 mk/$\sqrt{\mathrm{Hz}}$ in 200 nm in NDs at room temperature [18,19,20,21,22]. Furthermore, the narrow zero phonon line (ZPL) of silicon-vacancy (SiV) center has also shown a great promise as a single-photon source for quantum applications due to a high photon emission rate (short excited-state lifetime at room temperature) [23,24]. Other near-infrared color centers in diamonds have recently shown a promising application especially in quantum sensing applications, which could be a good candidate for such applications in the future [5,25].

Besides introducing a biocompatible SiV color center into BNDs, cleaning the surface of NDs from unwanted surface impurities, disordered carbons (graphite shells), and metal oxides that were introduced during growth and size-reducing processing is a critical step before the ultimate applications of BNDs. It was found that the best method to clean NDs is molten potassium nitrate (KNO_3_) treatment at a high temperature in order of 600 °C [26,27,28,29]. This etching treatment was shown to produce NDs with rounded shape and clean surfaces better than acid-cleaning and air oxidization methods. Furthermore, fluorescent NDs with rounded shapes showed significantly improved optical properties and excellent colloidal stability [15,26].

In this work, prior to silicon implantation into BNDs, we cleaned the surface of the BNDs and made them rounded using a well-established and efficient molten KNO_3_ treatment method. Size and morphology of the etched BNDs were characterized and compared to untreated BNDs. Silicon vacancies were incorporated into the rounded BNDs after appropriate ion implantations and post-annealing. Created SiV color centers in BNDs showed a good optical temperature sensing performance over a biological temperature range with good sensitivity. The obtained results hold a great potential for the promising multi-functions of the BND applications as anticancer agents in hyperthermia and thermoablation therapy.

## 2. Materials and Methods

### 2.1. Boron-Doped Nanodiamonds Etching in Molten Potassium Nitrate (KNO_3_)

Crushed boron-doped nanodiamonds (BNDs), ~0.5 wt% of boron, with average size of 100 nm used in this study were purchased from Adamas Nanotechnologies, Raleigh, NC, USA. To clean the surface of the purchased BNDs and smoothen their sharp edges to rounded, we used a well-known molten potassium nitrate (KNO_3_) treatment at a high temperature of 600 °C for 10 min as illustrated in Figure 1a [26,27]. For cleaning BNDs in potassium nitrate, 2 mg (corresponds to 400 µL of 5 mg/mL BNDs in DI water) of BNDs sample was mixed with 1 g of KNO_3_ powder and then mixed very well in an agate mortar. The homogeneous mixture was placed in a quartz boat and placed in a Thermolyne 21100 tube furnace and annealed at 600 °C for 10 min. The sample was then cooled down to room temperature, dissolved in water, and centrifuged. The pellet was resuspended in 2 mL of water, sonicated for 10 s in an ultrasonic bath, and centrifuged again. This washing step was repeated five times to completely remove salt residue. After that, the cleaned BNDs solution was placed in a glass vial and stored for further use. After appropriate cleaning of the BNDs, size, morphology, x-ray diffraction (XRD), and Raman measurements of the treated BNDs were performed to ensure that the cleaned NDs retain their structural and optical properties.

### 2.2. Raman Measurement of the Cleaned BNDs

Raman spectra measurements of the cleaned BNDs were performed on a commercial Raman system (Thunder optics, model TO-ERS-532, Montpellier, France) equipped with 100× air microscope objective and a continuous wave (CW) green laser (532 nm) with maximum power of 50 mW. The used green laser was a single mode with a narrow line width of 0.06 nm and diffraction limit spot size. The BNDs sample was drop-casted on a quartz substrate and excited with green illumination. The collected Raman spectra were then recorded and analyzed with a spectrometer unit attached to the Raman system.

### 2.3. Preparation of BNDs Samples for Confocal Imaging

The quartz slides used for sample measurements were first rinsed with acetone to remove any oils and dirt. After that, the slides were placed on a heating plate inside of a fume hood. Samples from the cleaned BNDs were then dropped onto the cleaned quartz slides using a transfer pipette and the temperature was raised to 200 °C. Once the samples were completely dry, they were placed in a tube furnace set to 550 °C and allowed to oxidize for 10 min. After 10 min, the samples were removed and cooled to room temperature before being placed on the confocal microscope for optical analysis before and after implantation and post-annealing in vacuum. Also, BNDs samples were then analyzed for Raman shift using green laser and for color centers’ investigation after implantation and required post-annealing. For comparison, as-received BNDs were cleaned in boiling acids. In this step, as-received BNDs were cleaned in a boiling mixture of nitric and sulfuric acids (HNO_3_:H_2_SO_4_ = 1:1), boiling at 120 °C for 4 h and washed with DI water through centrifugation steps. The acid-cleaning approach is known to etch graphitic carbon from ND surfaces. After that, the acid-cleaned BNDs were air oxidized at 600 °C for 10 min and then drop-casted on a quartz slide for temperature measurements.

### 2.4. Custom-Made Confocal Microscope for Optical Characterizations

To optically characterize the treated BNDs and perform optical temperature sensing measurements, a confocal laser scanning microscope was assembled as demonstrated in Figure 1b. The custom-built optical microscope was equipped with a near-infrared corrected microscope objective (100×, NA = 0.8, model number: LMPlanFL N, Olympus, Tokyo, Japan), a home-made spectrometer, photon counter, and a CW 690 nm red laser. The used red laser had a high stability and round beam (gaussian) with a minimum spectral error. To perform optical measurements of the cleaned BNDs samples, they were drop-casted on quartz substrates and then placed on the confocal microscope optical x–y–x stage. Optical scan of the cleaned BNDs was performed using a x–y scanner under a red (690 nm) laser with excitation intensity of 10 W/cm^2^. The obtained optical emission spectra from the SiV centers were collected by the same microscope objective and analyzed with a custom-made spectrometer with an 1800-grooves/mm diffraction grating (0.03 nm spectral resolution), sensitive camera, and a single photon counter.

### 2.5. Ion Implantation Process

The cleaned BNDs that showed promising diamond Raman line were prepared for ion implantation. So, several samples were irradiated at an irradiation commercial facility (CuttingEdge Ions, LLC, Anaheim, CA, USA). We irradiated the samples with silicon implantation at energy of 50 keV with a dose of 2 × 10^14^ ion/cm^2^. Post-annealing was required to mobilized the created vacancies in the implanted samples. For this, appropriate post-annealing was performed in vacuum at 1100 °C for 1 h to create SiV color centers in the cleaned BNDs.

### 2.6. Optical Temperature Sensing with the Cleaned and Irradiated BNDs

For the optical temperature measurements with SiV color centers in BNDs, a well-controlled and precise heating stage was placed on the confocal microscope. After focusing on the desired optical spots (NV centers) in the x–y scan, the heating stage was triggered to heat the BNDs sample over a specific temperature range, and the corresponding fluorescence from the SiV color center ZPL was recorded and then analyzed with a high-resolution spectrometer attached to the optical confocal microscope. Finally, the recorded ZPL shifts as a function of temperature increase and was plotted and analyzed.

## 3. Results and Discussion

Size and morphology of the clean BNDs were characterized and compared to untreated NDs using a transmission electron microscope (TEM). Figure 2a illustrates BNDs (as received) with irregular shape and uncleaned surfaces. On contrast, the KNO_3_-treated BNDs showed diamond nanocrystals with clean surfaces and rounded edges as depicted in low- and high-magnification TEM images in Figure 2b.

Figure 3a shows the average particle size of the treated BNDs that was then confirmed to be 100 nm using a dynamic light shattering system (DLS). This average particle size after KNO_3_ treatment showed a good agreement with particle size extracted from TEM images. Optical Raman measurement of the cleaned BNDs was carried out using a commercial Raman System equipped with green laser (532 nm) laser excitation. Figure 3b demonstrates a sharp and clean Raman spectrum of the BNDs with a peak at 1332 cm^−1^, which matches a diamond Raman peak recorded for a bulk diamond crystal [30,31]. The absence of amorphous carbon and graphite (D and G) peaks located at 1351 cm^−1^ and 1560 cm^−1^, respectively, indicates a clean surface of the BNDs after KNO_3_ treatment. Furthermore, the X-ray diffractogram (XRD) of the cleaned BNDs nanocrystals was performed and exhibits clear Bragg reflections at 43.9°, 74.9°, and 91° due to the cubic diamond (CD) of a lattice constant of a = 3.567 Å, which corresponds to (111), (220), and (311) in the diamond crystal lattice planes, as illustrated in Figure 3c. These index values are in good agreement with diamond XRD standard peaks reported in JCPDS card number 6-0675 [32].

In order to study and analyze color centers and perform optical temperature sensing using the treated BNDs, building a custom-built laser scanning confocal microscope was required. For this, we designed a confocal laser scanning microscope described Section 2. In this optical microscope, the sample is scanned with the desired laser excitation through the microscope objective, and the fluorescence is collected back through the same objective and forwarded to the detection unit, which consist of a single photon counter and a spectrometer. The excitation laser and other unwanted signal were perfectly filtered out from the fluorescence signal by optical filters.

Next, after cleaning and smoothing of the morphology of the BNDs, we decided to create silicon vacancies in the treated BNDs and test their optical performance prior to quantum temperature sensing experiment. The negatively charged SiV center in diamond is a point defect formed by the replacement of two adjacent carbon atoms with a single silicon atom as illustrated in Figure 4a. Upon a biocompatible laser excitation (mostly 690 nm), the SiV color center in diamond crystal emits a bright, narrowband, optical transition peaked at 737 nm (called the zero-phonon line ZPL) between its ground and excited states as illustrated in Figure 4b [17,19]. To experimentally create SiV color centers in the treated BNDs, appropriated ion implantation with silicon atoms at an implantation energy of 50 keV and doses of 2 × 10^14^ ion/cm^2^ was carried out followed by a required post-annealing in vacuum at 1100 °C for two hours in order to mobilize the created vacancies to form SiV center complex in the cleaned BNDs. Upon 690 nm laser excitation, the optical emission of the silicon-implanted BNDs showed a strong and narrow SiV center emission with its strong ZPL peaked at 737 nm, as shown in Figure 4c, which was in good agreement with SiV center emission observed in bulk diamonds implanted with silicon atoms [17,19]. The photostability of the detected optical emission from the SiV center was performed and revealed a stable emission over a thousand minuets under a continuous laser illumination. The density of SiV color centers within the BNDs nanocrystals are expected to be almost equal to the ion implantation dose density.

Optical temperature sensing using SiV color center created in the cleaned BNDs was performed over a small temperature range (296–310 K) within the region of interest for biological systems. For this, a droplet of 1 mg/mL of the treated BNDs was spin-coated on a piece of quartz to form well-dispersed SiV emitters. The sample was attached to a compact heater placed on the laser scanning microscope for optical temperature measurement using BNDs nanocrystals. On the heating stage, temperature was calibrated over a temperature range (298–330 K) in order to avoid possible laser heating confusion. After that, we scanned the BNDs to isolate small crystals with low counts close to single emitters counts. Figure 5a shows optical temperature sensing with SiV center as its ZPL peak shift to the red as the temperature increases. Figure 5b demonstrates SiV center ZPL shifts for two representative temperatures of 296 K and 310 K. Fitting the experimental data of optical temperature sensing of SiV color center in BNDs, shown in Figure 5a, to a linear equation, a slope of 0.014 nm/K is obtained, as demonstrated in Figure 5c, which provides the calibration for temperature measurements. These results are in good agreement with optical thermometry with SiV color center in diamond nanocrystals over a similar temperature range. Finally, to estimate temperature sensitivity of SiV centers in BNDs vs. measurement time, we measured the temperature sensitivity using Allan deviation of the ZPL peak position changes at 298 K as a function of integration time following a well-established procedure reported in [19]. For 10 s of integration time, the minimal detected temperature change was estimated to be of 0.2 K. The obtained sensitivity in this study matches the critical optical temperature sensing sensitivity of 0.5 K for integration time of 10 s, achieved in a previous report that is summarized in Table 1 [19,21,33].

Compared to previous reports in Table 1, the relatively good optical temperature sensitivity obtained in this study using boron-doped NDs can be attributed to substitutional nitrogen (P1) centers within the diamond nanocrystals acting as donors, increasing the Fermi level of the BNDs enough to stabilize the SiV charge state. Also, cleaning the BNDs surface from disordered graphitic materials and spin noise is known to enhance the optical properties of diamond color centers [26].

To explain the physics behind using SiV center in diamond in thermometry, the photoluminescence (PL) spectrum of the SiV reveals a strong peak at 737 nm at room temperature as shown in Figure 5a. This peak shifts with temperature due to the phonon mixing of the excited state [18]. Similar behavior of other color centers in diamond such as tin-vacancy (SnV) and germanium-vacancy (GeV) color centers was reported in the literature [18,25,34]. The observed ZPL shift of the SiV center as a function of temperature can be attributed to the electron–phonon interaction and is much faster than the spontaneous decay rate of the exited state over a wide range of temperatures, resulting in a strong temperature dependence of both the line width of the transition and a shift of its spectral position [18].

It was important to perform optical temperature sensing with uncleaned BNDs and compare their performance to the KNO_3_-treated BNDs. Similar to the KNO_3_-treated BNDs’ temperature measurements on the optical confocal microscope, we performed optical temperature sensing of air-oxidized BNDs (not treated with KNO_3_), which showed a temperature susceptibility (slope) of 0.022 nm/K. This obtained value was well below the susceptibility obtained from the treated BNDs. This could be attributed to graphitic shell strongly bonded into the diamond surface as well as trapped defects usually found in the sharp edges of crushed NDs. It was shown that diamond surface defects and noises can substantially reduce spin coherence and quantum yield of the color centers, resulting in poor quantum sensing [35,36].

Looking forward into the future, small and rounded BNDs with size less than 10 nm with stable SiV and NV color centers are needed for more exciting biological applications. Such small and high-quality BNDs can be produced by either etching high-quality NDs with an average size of 20–30 nm or by bottom-up growth approach reported in many recent experiments [8,37]. We anticipate this as most scientists especially in biology are interested in good quality and photostable small fluorescent nanodiamonds (FNDs) with size less than 10 nm, which can easily enter most of the biological tissue membranes to perform more critical functions such as drug delivery and sensitive bio-imaging [8,37].

Regarding the photoinduced thermal effect of laser excitation of the SiV, which could be an obstacle for useful applications, SiV diamond color center was carefully chosen in this study that can be excited and detected with the biological transparency window where overheating of biological tissues was minimized. This makes the real application of SiV center in BNDs more feasible especially in hyperthermia.

## 4. Conclusions

In this work, BNDs were carefully cleaned and made rounded following a molten potassium nitrate at high temperature approach. SiV color centers were introduced into the cleaned BNDs after an appropriate ion implantation and post-annealing in vacuum, required to mobilize created vacancies to form the desired SiV complex. Optical temperature sensing using SiV in BNDs was performed over a range of temperature within the biological temperature window (296–308 K) with a good sensitivity better than 0.2 K in 10 s integration time. These results illustrate that there are likely to be better applications of biocompatible BNDs doped with SiV centers in hyperthermia and thermoablation therapy. The results obtained in this study open the door for engineering small and rounded fluorescent nanodiamonds (doped with silicon vacancies) with a size of less than 10 nm that can easily enter most biological tissue membranes to perform more complicated tasks in biological applications.

## Figures and Tables

**Figure 1 materials-16-05942-f001:**
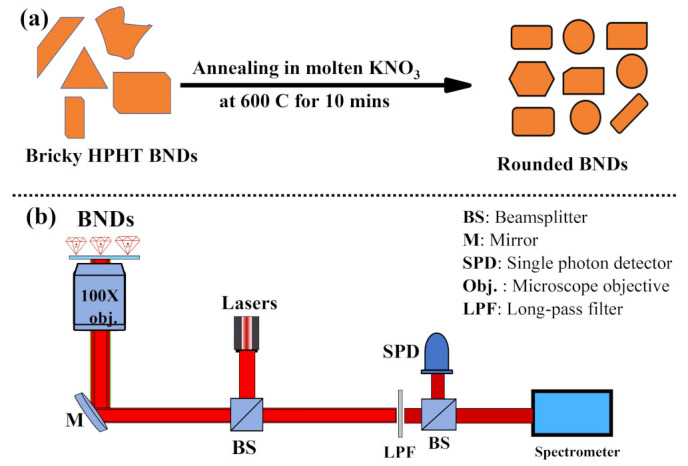
(**a**) An illustration of the concept of cleaning diamond nanocrystals with molten potassium nitrate (KNO_3_) at high temperature for a short period of time. This approach is expected to produce a cleaned and rounded diamond nanocrystals for enhanced optical temperature performance. (**b**) Experimental optical setup designed to study optical temperature sensing in BNDs. The optical setup has high numerical aperture microscope objective, galvo-scanner, single photon counter, and a high-resolution home-made spectrometer.

**Figure 2 materials-16-05942-f002:**
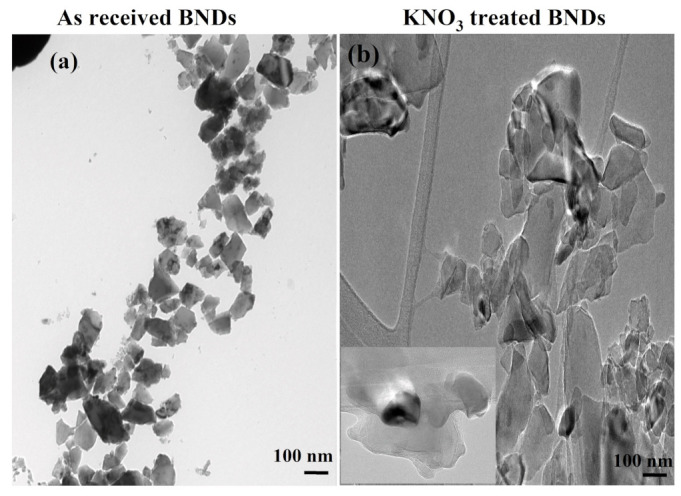
(**a**) TEM image of as-received BNDs shows uncleaned surfaces, irregular shapes, and sharp edges of BNDs. The average size of the received BNDs can be estimated from the TEM image to be in order of 100 nm. (**b**) The KNO_3_-treated BNDs illustrate clean surface and round-shaped crystals with average size of less than 100 nm. ((**b**), inset) shows a zoomed image of clean surface and rounded BNDs.

**Figure 3 materials-16-05942-f003:**
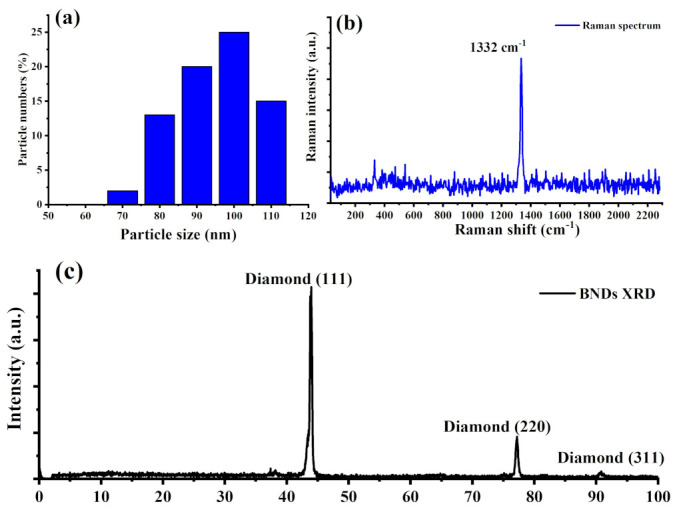
(**a**) Particle size of the cleaned BNDs nanocrystals cleaned by molten potassium nitrate (KNO_3_) at high temperature for 10 min collected using DLS system. The particle size obtained from the DLS is in good agreement with particle size obtained from TEM images. (**b**) Raman spectrum of the KNO_3_-treated BNDs reveals a strong diamond Raman peak at 1332 cm^−1^, which is in agreement with standard diamond Raman line in the literature. Interestingly, we observed no D and G peaks in the collected Raman spectrum of the cleaned BNDs, which indicate a clean surface of the diamond nanocrystals without graphite and amorphous carbons. (**c**) XRD pattern of the cleaned BNDs shows the cubic diamond diffraction patterns of a lattice constant of a = 3.567 Å, which corresponds to (111), (220), and (311) in the diamond crystal lattice planes.

**Figure 4 materials-16-05942-f004:**
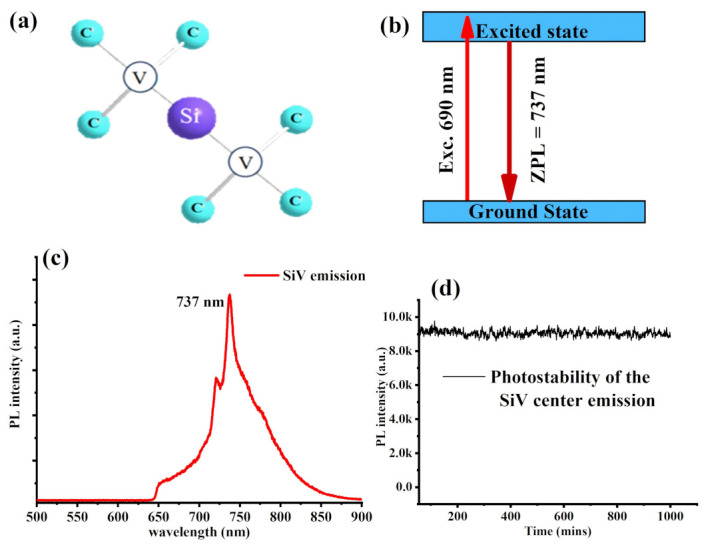
(**a**) A systematic illustration of creation of silicon vacancy in the diamond crystals. The SiV center in diamond is a point defect formed by the replacement of two adjacent carbon atoms with a single silicon atom. (**b**) An illustration of the electronic structure of the SiV in diamond, which consist of ground and excited states. Upon red excitation, the SiV center emits a bright, narrowband, optical transition peaked at 737 nm (called the zero-phonon line ZPL) between its ground and excited states. (**c**) Optical emission of the implanted NDs showed a strong and narrow SiV center emission with its distinguished ZPL peaked at 738 nm. (**d**) Shows a time stability of the optical emission recorded from the created SiV centers in BNDs.

**Figure 5 materials-16-05942-f005:**
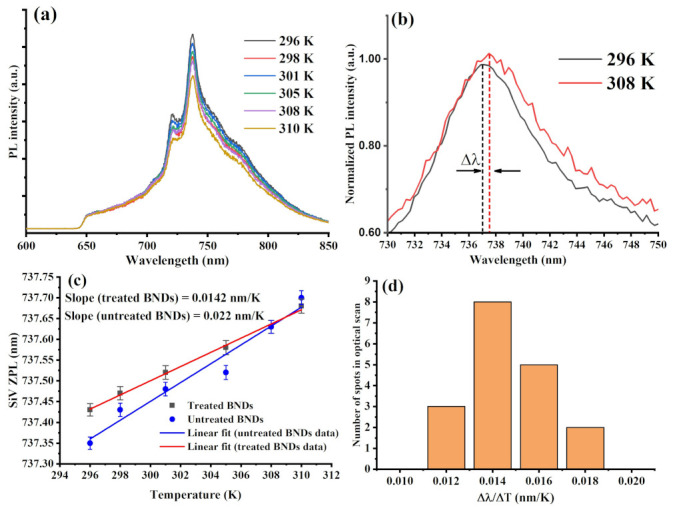
(**a**) Optical temperature sensing with SiV center as its photoluminescence (ZPL peak) exhibits a red shift as the temperature increases over a small range of temperature in the range of 296–308 K, which is of interest for biological systems. (**b**) Photoluminescence emission spectra of the SiV color center in diamond under 690 nm illumination as a function of different temperatures (296 K, black and 308 K, red). (**c**) A linear fitting of the SiV zero-phonon line (ZPL) position change as a function of temperature changes over a relatively wide range (296 K–310 K). (**d**) Shows a histogram of susceptibility (Δλ/ΔT) of the SiV center in BNDs for more than 15 measured bright spots in the optical scan. The average value is 0.0142 ± 0.002 nm/K. These optical spectra of the created SiV color center in BNDs were measured using a custom-made spectrometer with an 1800 gr/mm diffraction grating (0.03 nm spectral resolution).

**Table 1 materials-16-05942-t001:** Summary of optical temperature sensing with SiV color centers in NDs performed at experimental conditions similar to the current study.

Temperature Susceptibility (nm/K)	Temperature Sensitivity for 10 s	Temperature Range	Reference
0.016	0.5 K	295–313 K	19
0.013	NA	298–327 K	33
0.0124	0.24 K	298–308 K	21
0.012	0.2 K	296–308 K	This work

## Data Availability

The data presented in this study are available on request from the corresponding author.

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
