# Peer review of "Silicon Vacancy in Boron-Doped Nanodiamonds for Optical Temperature Sensing"

_materials, 2023, doi:10.3390/ma16175942_

Round 1

Reviewer 1 Report

I do encourage the authors to postpone their submission, and to strengthen their study, by including more investigations. Also, I would like to see the revised manuscript after serious major revisions before making a further decision on its publication in Materials.

The quality of English is alright

Author Response

Dear Reviewer,

Best regards,

Masfer

Reviewer 2 Report

There are a number of questions and comments on this article:

-It is unclear why there is a lack of citation of related articles by C. Nguyen, Appl. Phys. Lett. 112, 203102 (2018); S. Choi, ACS Photonics 6, 1387 (2019); T. Zhang, Journal of Materials Chemistry C, 10, 13734 (2022); and J. Zhang, Journal of Physical Chemistry C, 127, 3013 (2023).

-In the text, it is necessary to indicate the content of boron as well as SiV.

- There are no data on the spectral error and the diameter of the laser radiation spot in the focus in the method for describing the measurements of the Raman spectra.

-In the description of the optical characterization technique, there is no data on the red laser (pulsed or continuous, linewidth, power, divergence) and data on the spectral error.

-In Fig. 2b, on the y-axis should be the signature "Raman intensity". On Fig. 4c, it is necessary to add an error to the experimental points.

-According to Fig. 2a in this paper there are study of diamonds of various sizes. But in the present version the Raman spectrum is presented for only one diamond (and the size is not indicated). It is known that diamond has a second-order Raman spectrum, the shape of which depends on the crystal size (see V. S. Gorelik and A. Yu. Pyatyshev, Diamond and Related Materials 110, 108104 (2020)). Full Raman spectra of diamonds of all sizes should be given in the revised version.

- Most of the first paragraph of Section 3 should be moved to Section 2. Figure 2d should also be moved to Section 2.

- Captions for Figs. 1, 2, and 4 are partially merged onto the adjacent page, which is inconvenient.

-In some cases, it is necessary to correct the superscripts and subscripts. Line 242 should contain a link to Fig. 4b.

-In line 121, it is necessary to correct the number of grooves in the diffraction grating.

-In lines 68, 82, 142, 159, and 179, the name of KNO3 must be corrected.

-In line 251, you can assume the special symbol instead of "sqrt".

-Figures 3a and 3b do not take on anything new, so I suggest deleting them. What is the point of separately citing Fig. 3c if it is in Fig. 4a?

- I could not find in the text the citation of references 27 and 28.

Author Response

Dear Reviewer,

Best regards,

Masfer

Reviewer 3 Report

In this research work entitled “Silicon-Vacancy in Boron-doped Nanodiamonds for Optical Temperature Sensing”, the author implanted Si ion into boron-doped nanodiamonds and study its optical temperature sensing properties. This is an interesting topic for materials chemistry, near-infrared cellular imaging and semiconductor processing technology communities. The experiments were designed well. And the performance assessments were well performed. All of the main statements are supported by experimental analysis and arguments. The paper provided some useful information for the development and applications of new optical-temperature characteristics of the Si Ion implanted boron-doped nanodiamonds. However, I have some comments and concerns about the evaluation of this work, the details are given below.

1.Line 19: ZPL expansion is missing, Add ‘Zero Phonon lines (ZPL)’. 

2.Line 95: Mentioned the manufacturer and model of the Raman spectrometer used.

3.In the method section, the PL spectra measurement details are missing. Particularly, excited wavelength for each sample and recording range.  

4.Figure 1a: The TEM morphology of KNO3-treated BND shows a smoother surface than its counterpart. It is not a rounded/circular/sphere shape. Suggest to modify the proper illustration.

5.Figure 1c: it is not appropriate to present two scale bars in one picture. Suggest to modify it. It can understand the author’s point to show the same scale in two figures 1b and 1c but the author should remove the 50 nm scale bar.

6.Line 172: Typo error ‘amorphas’, It is amorphous. Should modify it.

7.Line 242-243: ‘Figure 2(b)’ This is not consistent with the following statement. The author should recheck and rewrite the sentences.

8.Line 248-251: Provide more details about the calculation/measurement of sensitivity, such as linear regression equation or standard thermal sensitivity equation correlated with applied wavelength.

9.Line-252-254: These sentences are not consistent with the mentioned Figure 3a. The author should recheck and rewrite the sentences.

10.The author provides the correlation between the temperature and the NIR wavelength. This was done by applying the temperature and capturing the corresponding emission spectra. But in reality, the light wave would apply and measure the corresponding photoinduced thermal effect at the local area. So how this present study can be helpful from the application point of view?  It would be appropriate if the author provide more details and add before the conclusion section.

11.Line 289-290: “Looking forward into the future, Small and rounded BNDs with size less than 10 nm with stable SiV and NV color centers are needed for more exciting biological applications.” The author mentioned about 10 nm sized BND, but in this work, the author described 100 nm BND materials. Can not understand the author’s point here. Suggest to rephrase it.  

No grammatical problems exist. The written language is understandable, but a few paragraphs still require some English polishing.

Author Response

Dear Reviewer,

Best regards,

Masfer

Round 2

Author Response

Dear reviewer,

Best regards,

Masfer

Reviewer 2 Report

The author has corrected most of the comments. But some still remain:

- It is a pity that the author of the article will not be able to present the dependence of the Raman spectra of diamonds on size.

-In lines 91 and 139, it is necessary to correct the subscript.

-In the caption to Fig. 5b, it is probably SiV, not SnV.

- In the section "Experimental Technique", it is said about the optical characterization of processed nanodiamonds using a red laser with a wavelength of 690 nm. In the caption to Fig. 5b, it refers to a green laser with a wavelength of 532 nm. It is necessary to eliminate the contradiction.

Author Response

Dear reviewer,

Best regards,

Masfer

Reviewer 3 Report

Congratulation. The author carefully addressed the raised concern.

No Issues

Author Response

Dear reviewer,

Thank you very much for your valuable time and efforts.

Best regards,

Masfer